# Revealing the Cryptic Diversity of Wood-Inhabiting *Auricularia* (Auriculariales, Basidiomycota) in Europe

Jiří Kout [1,*] and Fang Wu [2]

1 Department of Forest Protection and Entomology, Faculty of Forestry and Wood Sciences, Czech University of Life Sciences Prague, Kamýcká 1176, CZ-165 21 Prague, Czech Republic

2 Institute of Microbiology, School of Ecology and Nature Conservation, Beijing Forestry University, Beijing 100083, China; fangwubjfu2014@bjfu.edu.cn

* Correspondence: martial@seznam.cz

**Abstract:** Some unusual specimens of the wood-inhabiting fungus *Auricularia auricula-judae* have been studied using morphological and molecular methods. As expected from external features, we describe a new species *Auricularia cerrina* sp. nov. Sequencing of the ITS region confirms differences from other species of *Auricularia,* and preliminary phylogenetic analysis is presented. *Auricularia cerrina* is characterized by blackish fruitbodies in fresh conditions with the combined presence of the medulla layer and small spores compared with *Auricularia auricula-judae*. The new species is based on specimens from the Czech Republic (central Europe) of *Quercus cerris*. Colour photographs in situ of fruitbodies and some microscopic photos are provided.

**Keywords:** lignicolous fungi; oak forest; taxonomy; jelly fungi; heterobasidiomycetes

## 1. Introduction

A traditional group of jelly fungi (formerly included in Heterobasidiomycetes) has been revealed as polyphyletic [1] and includes some remarkable fungi with macroscopic fruitbodies or medicinal fungi, as well as inconspicuous species. Ear-shaped fungi of the genus *Auricularia* Bull. (Auriculariaceae, Auriculariales, Agaricomycetes) are one of the most well-known with a cosmopolitan distribution. People like them as edible mushrooms, especially in Asia, where *Auricularia* fungi have been collected for a long time for their usage in traditional medicine [2]. The medicinal potential of *Auricularia* was confirmed by modern studies [3].

The genus *Auricularia* is characterized by mainly pileate basidiocarps with a gelatinous consistency, brownish colours, with a smooth, pruinose to hairy, abhymenial upper surface, smooth hymenophore (except "merulioid" species *Auricularia delicata* (Mont. ex Fr.) Henn. complex), three times transversally septate basidia with allantoid, inamyloid, hyaline basidiospores, saprotrophy (white rot), or weak parasitism of woods [4,5]. The jelly-like consistency of the fruitbodies and its auricularioid shape generally enable the easy identification of the genus.

The taxonomy of *Auricularia* has been based on morphology for a long time [4–6]. Molecular data confirm the genus as monophyletic [7–9]. However, new cryptic species revealed over time [10–12], are often unrecognisable through external morphology [13,14]. The taxonomy of the genus has thus made significant progress, and some species are becoming quite difficult to identify due to the morphological variability of fruitbodies at different stages of development. Exact identifications of *Auricularia* species are based on some microscopic features (dimensions of spores and basidia, length of abhymenial hairs), DNA sequences, and biogeographical patterns [11–14].

In Europe, auricularioid fungi have been considered two species for a long time— *Auricularia auricula-judae* (Bull.) Quél. and *Auricularia mesenterica* (Dicks.) Pers. [15–17].

Both species of *Auricularia* have never been generally of main interest to European mycologists due to their seemingly clear taxonomy. Semi-resupinate *A. mesenterica* is recorded from different hardwoods. Pileate *A. auricula-judae* has been presented as a species often growing on elderberry (*Sambucus* sp.), nevertheless, its host range is becoming wider, probably due to climate change [18]. *Auricularia auricula-judae* was marked as widely spreading in temperate parts of the Northern Hemisphere [5,6]. Duncan and Macdonald [19] examined this species throughout Europe and North America (hardwood and conifer substrates) using cultural studies and revealed an incompatibility between continents and different substrates as well. Subsequently, modern molecular analysis confirms the existence of cryptic species within *A. auricula-judae* [13] and biogeographic areal of *A. auricula-judae* is going to become more limited only for Europe [12,20–22].

The aim of this article is to present a new species of *Auricularia* from Europe. Selected collections of *A. auricula-judae* were morphologically and molecularly analysed. The received results confirm preliminary expectations for unusual specimens collected from central Europe (area relatively well-explored in fungi). Therefore, a new *Auricularia* species is described in the present study.

## 2. Materials and Methods

### 2.1. Collection of Specimens, Morphological Methods

The material of several species of *Auricularia* was collected and examined in the field. Some of them were selected for detailed morphological examinations. The study is aimed at the *A. auricula-judae* (12 specimens from different substrates and different countries), three specimens moreover were separated as a new species (holotype, paratypes). Two specimens one of *Auricularia cornea* Ehrenb. and another of *Auricularia fuscosuccinea* (Mont.) Henn. were used for a basic comparison because we need some species with a medulla layer; in addition, the traditional "tropical" species *A. cornea* was recorded in Europe [14].

A new species is described from the Rendezvous National Nature Monument near Valtice in the Czech Republic (Figure 1). The type locality is covered by an oak (*Quercus* spp.) with thermophilous forest (*Quercetea pubescentis*) as a main biotope.

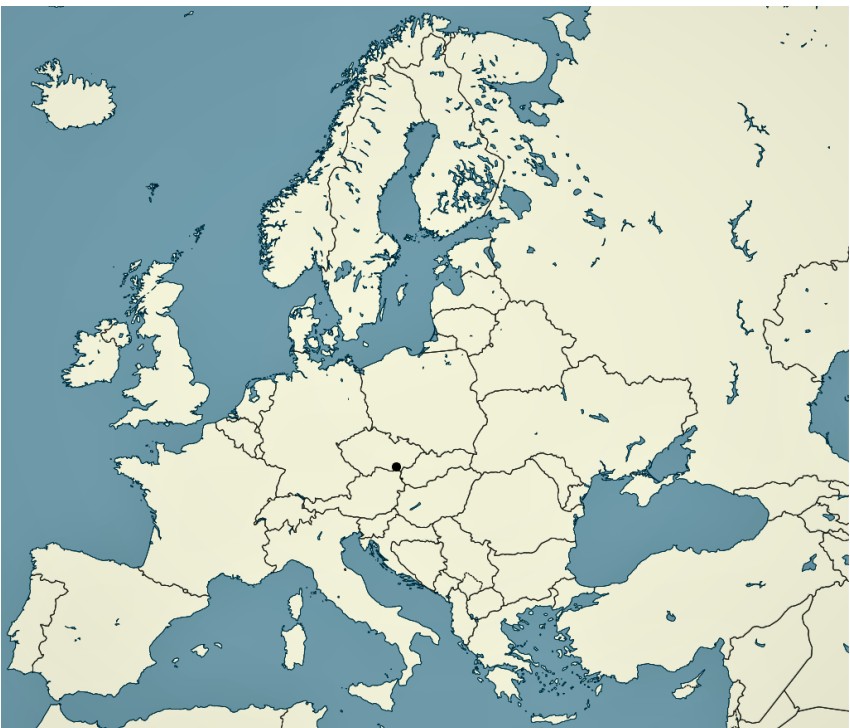

**Figure 1.** Location of the type locality of *A. cerrina* in the Czech Republic (black dot).

Specimens were studied in fresh conditions and in the laboratory under a microscope. Some fruitbodies of the collected material have been deposited in the public herbaria, some in the Mycological Herbarium of the Department of Biology, Geosciences and Environmental Education, the University of West Bohemia (abbreviated UPL here) and in the personal herbarium of J. Kalián (marked as K). The inspections of some herbaria (BRNM, PRM, TFC) and studies of deposited specimens were also included. Abbreviations of public herbaria follow Index Herbariorum (http://sweetgum.nybg.org/ih/) accessed on 18 February 2022 [23].

Specimens examined (holotype and paratypes specimens are mentioned by the description of species):

*Auricularia auricula-judae*. Czech Republic. Near Plzeň, on *Sambucus nigra*, 29 May 2014, J. Kalián (herb. J. Kalián K00672014); Plzeň, on *Koelreuteria paniculate*, 7 June 2016, J. Kout (UPL); Plzeň, on *Laburnum anagyroides*, 12 August 2017, J. Kout (UPL); near Plzeň, on *Populus tremula*, 11 August 2019, J. Kout (UPL). Near Křivoklát, on *Abies alba*, 28 October 2011, J. Kout (JK 1110), GenBank OM747869. Near České Budějovice, Vrbenské rybníky NNR, *Salix* sp., 11 December 2005, J. Kout (JK 0512, UPL). Ústí nad Labem, on *Sambucus nigra*, 24 December 2006, L. Zíbarová (LZ 0612); Kokořínsko, Mokřady dolní Liběchovky National Reserve, on *Sambucus nigra*, 24 April 2018, L. Zíbarová (LZ 1804). France. Dépt. Alpes Maritimes, on *Hedera helix*, 21 September 1998, leg. D. Triebel et G. Rambold, det. P. Roberts (PRM 901343, D. Triebel: Microfungi Exsiccati 493). Spain. Canary Islands, Tenerife, Vueltas de Taganana, on *Ocotea foetens*?, 7 May 1972, W. Wildpret et E. Beltrán-Tejera (TFCMic 00055, duplic. UPL); Gran Canaria, Los Tiles de Moya, on *Ocotea foetens*?, 16 April 1973, E. Beltrán-Tejera (TFCMic 00056, duplic. PRM 956932, UPL, GenBank OM747870); La Palma, Los Tilos, humid lauroid forest, on hardwood, 7.4. 1972, E. Beltrán-Tejera (TFCMic 00157, duplic. UPL).

*Auricularia cornea*. Indonesia. Bali, 21 August 2017, M. Jarošová (MJ 1708 UPL), GenBank OM747873.

*Auricularia fuscosuccinea*. Venezuela. Cueva del Guacháro, 26 February 2004, J. Kout (UPL).

Macroscopic characters were observed on fresh and dry material by a stereomicroscope Olympus SZ51. Microscopic features were examined in water, 5% KOH solution, Melzer's solution, Cotton Blue in lactic acid, and Congo-red in L4 [24] using an Olympus BX 51 light microscope. The spores were studied in Congo-red in a 5% ammonia solution and Cotton Blue solution. The spore measurements are based on 20 well-developed spores from sections of fruitbodies at 1000× magnification under an oil immersion lens.

The internal structure of fruitbodies in *Auricularia* is divided into several layers [4]. However, the dimensions of these zones were not found to be very valuable features [25], the variability is too high. Phylogenetic studies generally do not use it [13,14], except when the presence or absence of the medulla layer and the length of hairs on pileus surface. The presented work accepts it in the same way, thanks to our experiences with microscopic work with the thickness of the sections, which significantly affects the dimensions of internal zones in fruitbodies.

## 2.2. Molecular Methods

A small piece of dried fruitbody (approx. 0.3 g) was frozen and then crushed for 60 s with a steel ball in a mixer mill MM301 RETSCH under liquid nitrogen. Subsequently, the obtained material was processed for isolation of total DNA using the CTAB/NaCl method as described by Murray and Thompson [26], followed by repeated extractions with chloroform and isopropanol precipitation. Isolated DNA was dissolved in 100 μL of sterile water and purified using the Wizard Clean Up kit PROMEGA. The resulting DNA solution (50 μL) was diluted 10 times and 1 μL was used as a template for amplification of the internal transcribed spacers (ITS). The PCR method was utilised according to Vlasák et al. [27]. The ITS region was amplified with ITS5 and ITS4 primers [28]. Amplified DNA was purified using the Wizard Clean Up kit PROMEGA and sequenced in the Eurofins Genomics. We used

sequences of the internal transcribed spacers for molecular determination because it was marked as a suitable and sufficient part of DNA for *Auricularia* species resolution [13].

There are listed specimens selected for phylogenetic analysis in Table 1. Some sequences of the nuclear ribosomal ITS were retrieved from GenBank and five sequences were generated newly from our specimens. Non-auricularioid *Elmerina efibulata* (Y.C. Dai and Y.L. Wei) Y.C. Dai and L.W. Zhou from Auriculariales was chosen as outgroup.

**Table 1.** Specimens selected for phylogenetic analysis (newly generated sequences in bold).

| Species | Sample | ITS number |
|---|---|---|
| *Auricularia africana* | T3 | MH213349 |
| *A. africana* | Ryvarden 44929 | MH213350 |
| *A. americana* | Dai 13476 | KM396766 |
| *A. americana* | LE 296428 | KJ698429 |
| *A. americana* | HHB 11370 | KM396766 |
| *A. angiospermarum* | TJV-93-12-SP | KT152096 |
| *A. angiospermarum* | Cui 12360 | KT152097 |
| *A. angiospermarum* | HHB 11037 | KT152098 |
| *A. asiatica* | Dai 16149 | KX022010 |
| *A. asiatica* | OM 13932 | MZ618931 |
| *A. asiatica* | BBH 895 | KX621160 |
| *A. auricula-judae* | MW 446 | AF291268 |
| *A. auricula-judae* | Dai 13210 | KM396769 |
| *A. auricula-judae* | MT 7 | KM396771 |
| ***A. auricula-judae*** | **JK 1110** | OM747869 |
| ***A. auricula-judae*** | **TFCMic 00056** | OM747870 |
| *A. australiana* | HN 3213 | MZ647504 |
| *A. australiana* | HT 190 | MZ647503 |
| *A. brasiliana* | CRSL 886 | KP729274 |
| *A. brasiliana* | AN-MA 42 | KP729275 |
| *A. brasiliana* | RSC 359 | KP729276 |
| ***A. cerrina* sp. nov.** | **HK 677** | OM747871 |
| ***A. cerrina* sp. nov.** | **HK 1510** | OM747872 |
| *A. conferta* | Dai 18825 | MZ647500 |
| *A. conferta* | Dai 18826 | MZ647505 |
| *A. cornea* | Dai 12587 | KX022012 |
| *A. cornea* | Dai 15336 | KX022014 |
| *A. cornea* | YG-DR 1 | MH213353 |
| ***A. cornea*** | **MJ 1708** | OM747873 |
| *A. fibrillifera* | Dai 13598A | KP765615 |
| *A. fibrillifera* | F 234519 | KP765610 |
| *A. fibrillifera* | Cui 6704 | KP765613 |
| *A. fuscosuccinea* | FP-102573 | KX022027 |
| *A. fuscosuccinea* | Dai 17451 | MH213368 |
| *A. fuscosuccinea* | Dai 17422 | MH213367 |
| *A. heimuer* | Dai 13503 | KM396789 |
| *A. heimuer* | Dai 13765 | KM396793 |
| *A. heimuer* | Dai 2291 | KM396785 |
| *A. lateralis* | Dai 16416 | KX022023 |
| *A. lateralis* | Dai 16420 | KX022025 |
| *A. lateralis* | Dai 15670 | KX022022 |
| *A. mesenterica* | LYBR 5353 | KM396801 |
| *A. mesenterica* | Kytovuori-89-333 | KP729284 |
| *A. mesenterica* | Miettinen 12680 | KP729286 |
| *A. mesenterica* | YG 029 | MZ618938 |
| *A. minutissima* | Dai 14881 | KT152104 |
| *A. minutissima* | Dai 15455 | KX022030 |
| *A. minutissima* | LE 296424 | KJ698434 |
| *A. nigricans* | Ahti 55718 | MH213372 |

**Table 1.** *Cont.*

| Species | Sample | ITS number |
|---|---|---|
| *A. nigricans* | TJY 93-242 | KM396803 |
| *A. nigricans* | Ahti 36234 | KM396802 |
| *A. novozealandica* | PDD 75110 | KX022032 |
| *A. novozealandica* | PDD 83897 | KX022034 |
| *A. novozealandica* | PDD 81195 | KX022033 |
| *A. orientalis* | Dai 14875 | KP729270 |
| *A. orientalis* | Dai 1831 | KP729271 |
| *A. orientalis* | Dai15813 | KX022036 |
| *A. pilosa* | LWZ20190421-7 | MZ647506 |
| *A. pilosa* | JMH 45 | KM267731 |
| *A. pusio* | AK 174 | MH213373 |
| *A. pusio* | AK 547 | MH213374 |
| *A. pusio* | Smith 18 | MH213375 |
| *A. scissa* | DR 777 | KM396804 |
| *A. scissa* | Ahti 49388 | KM396805 |
| *A. scissa* | TFB 11193 | JX065160 |
| *A. sinodelicata* | Cui 8596 | MH213376 |
| *A. sinodelicata* | Dai 13926 | MH213379 |
| *A. sinodelicata* | Dai 12242 | MH213381 |
| *A. srilankensis* | Dai 19522 | MZ647501 |
| *A. srilankensis* | Dai 19519 | MZ647507 |
| *A. srilankensis* | Dai 19575 | MZ647502 |
| *A. subglabra* | Dai 17403 | MH213382 |
| *A. subglabra* | Dai 17394 | MH213383 |
| *A. subglabra* | TFB 10046 | JX524199 |
| *A. submesenterica* | Dai 792 | KX022037 |
| *A. submesenterica* | Dai 14773 | KX022038 |
| *A. submesenterica* | Dai 15450 | MH213386 |
| *A. thailandica* | MFLU 130396 | KR336690 |
| *A. thailandica* | Dai 15335 | KX022026 |
| *A. thailandica* | Dai 13655A | KP765619 |
| *A. tibetica* | Cui 12267 | KT152106 |
| *A. tibetica* | Cui 12337 | KT152108 |
| *A. tibetica* | Dai 15604 | MH213388 |
| *A. tremellosa* | AJS 5896 | JX065162 |
| *A. tremellosa* | TENN 28734 | JX065159 |
| *A. tremellosa* | Dai 17419 | MH213391 |
| *A. villosula* | Dai 13450 | KM396812 |
| *A. villosula* | LE 296422 | NR137873 |
| *A. villosula* | MFLU 162128 | KX621164 |
| *Elmerina efibulata* | Yuan 4525 | MZ618945 |

The phylogenetic tree is constructed by Maximum Likelihood (ML) and Bayesian Inference (BI) methods based on an aligned length of 596 characters (all positions containing gaps were treated as missing data). The BI tree is presented here (Figure 2). The alignment is aligned using MAFFT 7.0 with the Q-INS-I strategy using the default parameters [29]. ML and BI methods were used for the ITS dataset to enhance the reliability of phylogenetic analyses. The optimal substitution models suitable for the dataset were determined using the Akaike information criterion (AIC) implemented in MrModeltest 2.3. [30]. The ML tree was constructed using RaxML HPC BlackBox running on XSEDE through the CIPRES Science Gateway-web (portal/platform, https://www.phylo.org accessed on 9 February 2022) and the BI tree was calculated with MrBayes3.2.5 under the GTR+I+G model [31].

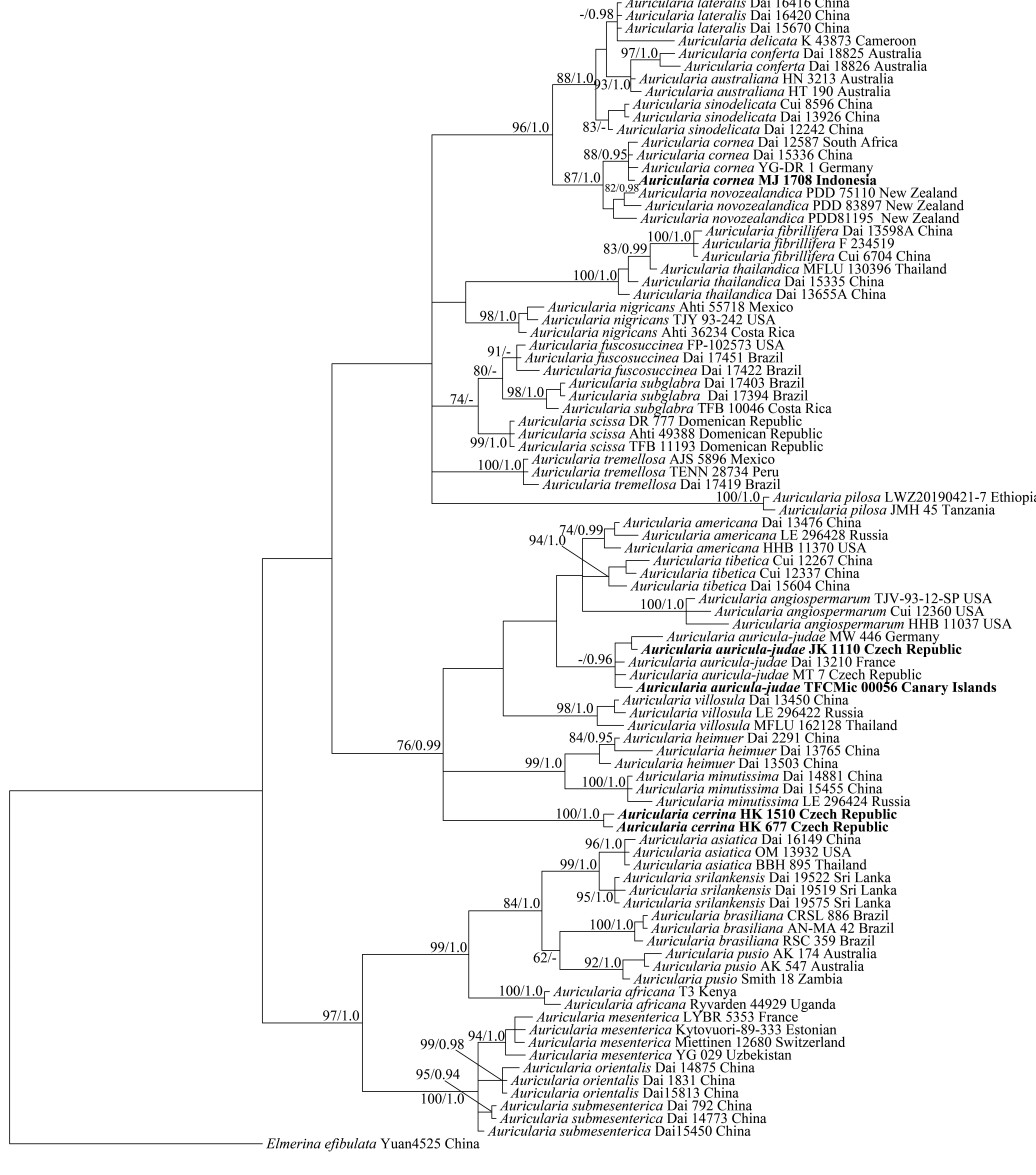

**Figure 2.** Bayesian phylogenetic tree (ITS region) of the selected specimens of *Auricularia*. Sequences generated in this study are in bold. Maximum likelihood bootstrap support and Bayesian posterior probability are shown.

## 3. Results

The ITS dataset included sequences from 91 fungal samples representing 32 species and had an aligned length of 597 characters. BI resulted in the nearly congruent topology with ML analysis, with an average standard deviation of split frequencies = 0.009892. Only the topology from BI analysis was present with values from BI and ML at the nodes. The phylogeny (Figure 2) demonstrated that the new species formed a distinct lineage with high support.

Auricularia cerrina Kout, Hejl and Kalián, sp. nov. Figures 3 and 4

MycoBank no.: MB 843069.

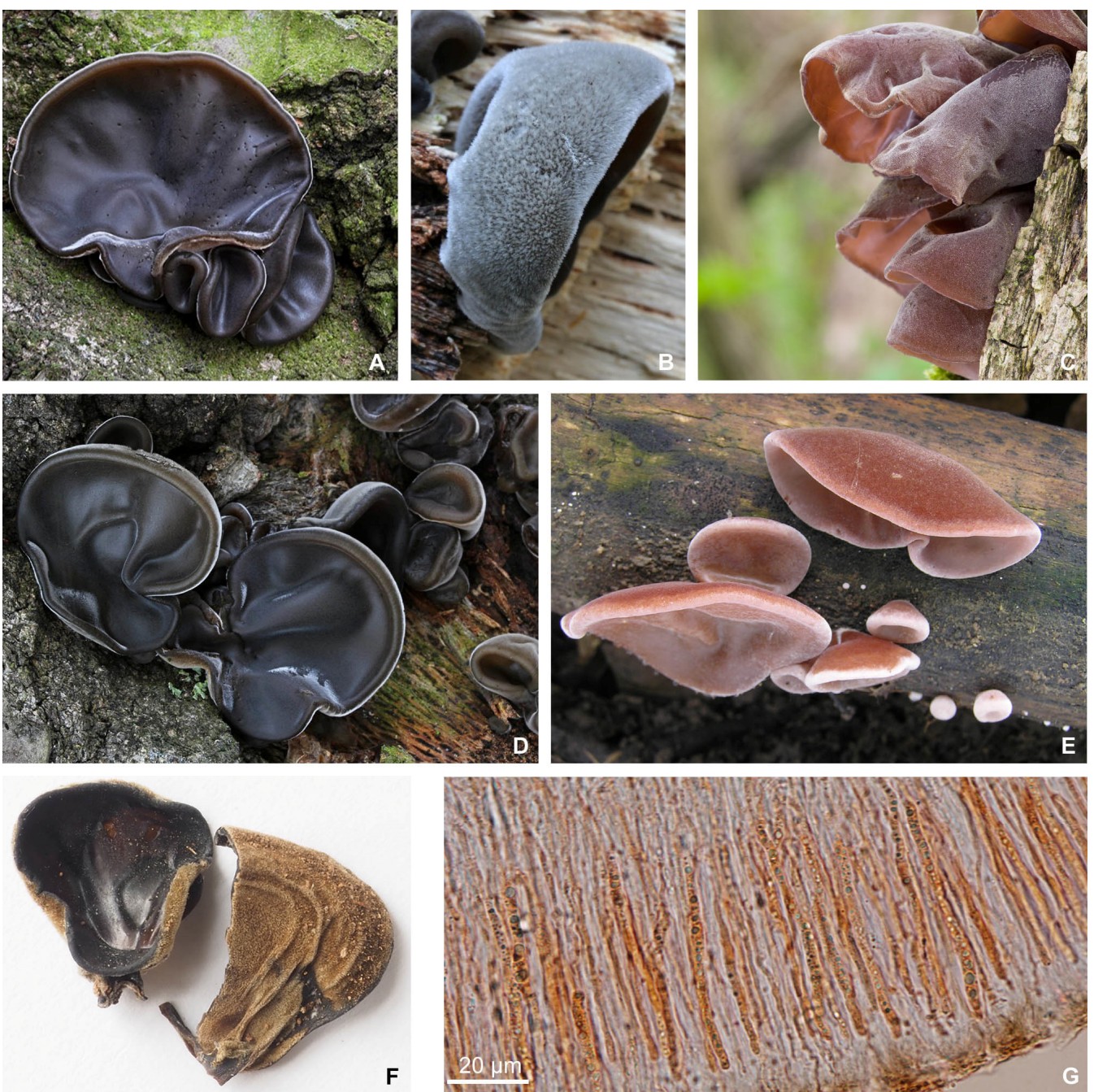

**Figure 3.** Comparison of *Auricularia* species: (**A,B,D**) Basidiocarps of *A. cerrina* in field (holotype), photo L. Hejl; (**C**) *A. auricula-judae* in field (LZ 1804), photo L. Zíbarová; (**E**) *A. auricula-judae* in field (LZ 0612), photo L. Zíbarová; (**F**) Dried specimen of *A. cerrina* (part of holotype); (**G**) Hymenium of *A. cerrina* (holotype).

Basidiocarps gelatinous in a fresh condition, begin as slightly cupulate and develop into an auricularioid shape, sometimes slightly retracted to the centre, attached in one point to the substrate, single or caespitose, projecting up to 40 mm in diametre and 1–4 mm in thick, thinner on the margins, dark greyish. Abhymenial surface azonate, dense, hirsute, hairs single or arranged in tufts, thick-walled along the entire length attenuated under tips, with visible lumen, apically obtuse, without a remarkable swollen base, only slightly widened, smooth, hyaline, occasionally pigmented on the base, 125–250 μm long, 5–7 μm in diameter, up to 11 μm width on the base, forming a whitish to greyish hairy layer that is

remarkably in contrast to the dark-coloured surface of the pileus. Hymenophore smooth, sometimes with occasional folds, greyish to grey with brownish tint on the margin in a fresh condition, becoming black by drying. Hyphal system monomitic, hyphae with clamp connections, branched, interwoven, smooth, hyaline. Medulla presents in context, consisting of parallel hyphae 2–4 μm wide. Scattered crystals are present in the hymenium. Basidia cylindrical with three transverse septa, (49–)53–65 × (4.5–)5–5.5 μm. Basidiospores allantoid, smooth, hyaline, with drops, 11.5–16 × (4–)5–6 μm, negative in Melzer's solution.

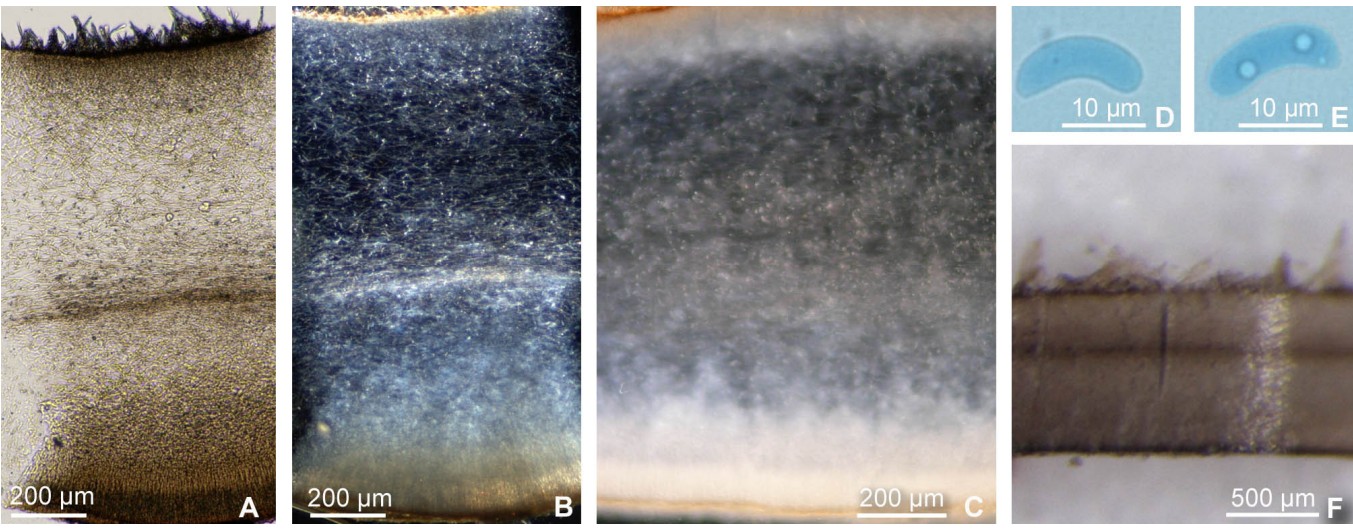

**Figure 4.** Microscopic features of *Auricularia* species: (**A,B**) Vertical section of basidiocarp *A. cerrina* (holotype) with visible medulla; (**C**) Vertical section of basidiocarp *A. auricula-judae* (JK 0512); (**D,E**) Basidiospores of *A. cerrina* (paratype HK 1510); (**F**) Medulla of *A. cerrina* (holotype) visible under stereomicroscope (dried specimen).

Holotype: Czech Republic. Břeclav District, near Valtice, Rendezvous National Nature Monument (48°44′53.780″ N, 16°47′31.863″ E), 200 m a.s.l., on lying branches of *Quercus cerris*, 25 October 2015, L. Hejl et J. Kalián (HK 677), PRM 956933, GenBank OM747871. Isotype UPL, K.

Paratype: ibid. 29 October 2015 (HK 1510, UPL, K), GenBank OM747872. Ibid., 19 July 2020, J. Walter (UPL).

Etymology—referring to the species name of the substrate tree.

## 4. Discussion

The description of *A. cerrina* is based on morphological characters and phylogenetic analysis. It is characterised by dark-coloured fruitbodies, grey to blackish in colour with dense whitish to grey hirsute pileus. Microscopically, there is a distinct medullary layer visible in section by eye and under a stereomicroscope and smaller spores compared to sympatric *A. auricula-judae*. The phylogenetic tree shows *A. cerrina* as a monophyletic clade in the *A. auricula-judae* species clade with 100% support, so the ITS region confirms it is sufficient to support a new species same as in previous studies [13,14,32]. It is an important confirmation, together with biogeographical areal, due to partial overlapping of morphological features in some species [14,32]. Widespread *A. auricula-judae* in the Czech Republic [17] may be confused with *A. cerrina* because nobody expected another species of *Auricularia* in well-explored Europe. Moreover, some dried specimens look similar and it is unlikely that anyone is performing a microscopic examination of the seemingly well-known *Auricularia*.

The type locality of *A. cerrina* and its known spreading is limited only at the biotope of a thermophilous oak forest (*Quercus cerris* as a substrate all collected specimens). If there is not a wider demonstrated distribution of *A. cerrina*, as well as its wider substrate preference,

then it may be considered an endangered species in the Czech Republic. Preliminary findings suggest that *A. cerrina* is European southern species from hardwoods (still limited to oak).

For the second time, we collected probably the same material from Greece (not from *Quercus*), basic morphological features agree, however, there are no more spores, and it was not sequenced. Similarly, Kobayasi [6] suggests that perhaps the same species is also described in a specimen from Greece. However, there is nothing known about spores and features in a fresh condition, in addition, the medulla is absent. During inspections of the herbarium, some specimens were revealed that could be *A. cerrina* (e.g., PRM 843445 from Slovakia and from *Q. cerris*). However, they are too old or without spores, so we also do not know their appearance in a fresh condition, and, therefore, they were not included in the description of the new species.

A well-known *A. auricula-judae* is an abundant temperate European species and it is currently the only species that can be mistaken for *A. cerrina* due to the common area of spreading. *Auricularia auricula-judae* is mainly recorded from *Sambucus nigra* [16] but it inhabits many species of woody plants (specimens examined, [14]). It was considered a widespread species before the introduction of molecular methods into taxonomy [5]. However, phylogenetic analyses revealed that it was a complex of species, and *A. auricula-judae* remained confined to Europe [13,14]. The substrate ecology of the species has developed in the opposite way. *Auricularia auricula-judae* was stated as a preferential species for elderberry in the past [15] and today it is regularly recorded on a wide range of species, deciduous woody plants to conifers. Conifers and hardwoods as a substrate are crucial points in the determination key in Wu et al. [13], where *A. auricula-judae* is situated under hardwoods. However, the examined specimen from *Abies alba* (JK 1110) was confirmed by sequencing as *A. auricula–judae* (the wood was inhabited by *Hymenochaete cruenta* (Pers.) Donk, specific species for *Abies*). Similarly, the length of the abhymenial hairs has an important determination position [13]. Several authors present abhymenial hairs from *A. auricula-judae* only up to 150 μm [6,7,13], however, it is possible to observe more variable lengths in European specimens (there are common hairs over 200 μm). Hirsute pileus may be influenced by climatic conditions as protection against desiccation, and then this feature may not be so stable.

There are several species of *Auricularia* similar to *A. cerrina*. They share some common features, such as smaller spores, presence of medulla, and the oak as substrate. These species as *Auricularia angiospermarum* Y.C. Dai, F. Wu and D.W. Li, *Auricularia heimuer* F. Wu, B.K. Cui and Y.C. Dai, *Auricularia minutissima* Y.C. Dai, F. Wu and Malysheva have limited spread outside Europe and differ in external features of fruitbodies and in case of ambiguity, the ITS sequence confirms the difference of *A. cerrina* [12–14].

**Author Contributions:** Conceptualization, J.K.; methodology, J.K.; investigation, J.K.; molecular analysis, F.W.; writing—original draft preparation, J.K.; writing—review and editing, F.W. and J.K. All authors have read and agreed to the published version of the manuscript.

**Funding:** This research received no external funding.

**Institutional Review Board Statement:** Not applicable.

**Informed Consent Statement:** Not applicable.

**Data Availability Statement:** Not applicable.

**Acknowledgments:** We are thankful to L. Hejl (Mykologický klub Nezvěstice) and J. Kalián (Mladý mykolog z.s.) for the collection of type material, J. Walter (Museum of West Bohemia in Pilsen) for additional specimens, similarly M. Jarošová and L. Zíbarová (some photos in addition). We would like to thank J. Vlasák (Biology Centre of the Academy of Sciences of the Czech Republic) for the isolation of the DNA, curators of inspected herbaria (BRNM, PRM, TFC) and M. Mergl (University of West Bohemia) for the arrangement of photos.

**Conflicts of Interest:** The authors declare no conflict of interest.

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
