# Peer review of "Revealing the Cryptic Diversity of Wood-Inhabiting Auricularia (Auriculariales, Basidiomycota) in Europe"

_forests, doi:10.3390/f13040532_

Round 1
Reviewer 1 Report
There must be a space after Table 1 to make it clear that the paragraph on phylogenetic analysis begins.
Figure 1. move to the Results section.
The Results are limited to the very detailed and adequately described sections on Taxonomy and Morphology.
There is no paragraph about ITS dataset and also the results of the phylogenetic analysis are missing. This needs to be added.
Because a new species is described in the study, it is recommended to supplement the analysis with another gene (n LSU or RPB2). Many of the articles you use in the Introduction prove it.
Line 48 dot missing after A. mesenterica
Line 192, 197, 214 follow the instructions for writing, change the beginning of the paragraph
Author Response
Dear reviewer,
we checked your comments and responses are in attachament.
Thank you
J. Kout, F. Wu

Reviewer 2 Report
Dear authors, I have read with interest the manuscript entitled "Reveal cryptic diversity of wood-inhabiting Auricularia (Auriculariales, Basidiomycota) in Europe".
Your research is interesting and important for the better understanding of forest fungal component. There are some suggestions that I consider will improve your work.
General comments - remove "we" and any personal expressions or words. Make the text impersonal.
The abstract should be rewritten in order to present point by point the findings in your manuscript.
Introduction
Line 41-43 - give the name of the microscopic features.
Line 56-59 - rewrite this paragraph. Here you need to present the aim of your study. The hypotheses or specific objectives, each of them in a single sentence. Give details on the collection, here or in Mat and Meth section.
Material and Methods
How many samples and species. Describe the area, maybe a map will help. Give some indications regarding vegetation type in the area. Insert a reference to Table 1 which contain some of the requested information.
I suggest you to move Figure 1 to Results section and make an explanation for it. This will sustain the next results.
Results section
Add few more sentences to make a smoother passage to the description of your species.
Something like The species Auricularia cerrina Kout, Hejl & Kalián, sp. nov., MycoBank no.: MB 843069, is described visually i figures 2 and 3.
The species present basidiocarps gelatinous in fresh.....
Discussion section
I suggest you to not make refrence to previous presented tables of figures from the results section. Remove (Figure 3F) and (Figure 1) from the text.
Change the map with one that present the area of specimen collection and move it to Material and Method section.
Pay attention to the alignment of your text, especially in the discussion section.
Rewrite the text between lines 187-196 in a more impersonal manner. It sounds like your own observations. Add more references in the discussion section, if possible.
Overall, the article is interesting, but it deserves an improvement in order to be clearer and easier to read.
Author Response
Dear reviewer,
we checked your comments, responses are in attachament.
Thank you
J. Kout, F. Wu

Round 2
Reviewer 1 Report
My comment to add another region aimed to increase the relevance of the results presented. ITS can be used alone or in conjunction with other protein coding genes for species identification. The LSU region when combined with the ITS region, can also be valuable for species identification in fungi and can be used in phylogenetic analyses as well (Raja HA et al. 2017 J. Nat. Prod. 80:756-770).
The authors emphasized that only the ITS sequence was sufficient to identify the species and added citations 13, 14 and 31 to their statement. I did not get a sufficiently clear answer as to why, according to the authors, only one gene region is sufficient. Published works (12, 13, 14, 31) was based on the analysis of a more than ITS gene region (see below).
In the paper 13:
„ ITS sequence data is a sensitive marker to discriminate species“. was written in Abstract . However, in Results we can find : „The phylogeny inferred from ITS+nLSU+rpb2 sequence data also demonstrated that the three new species, A. angiospermarum, A. minutissima, and A. tibetica, separate into three distinct clades with significantly higher support (80 %BS, 100 % BP, 0.95BPP; 100 %BS, 99 % BP, 1.00BPP; 97 %BS, 100 % BP, 0.97BPP) than those from ITS sequences alone.
- Wu F., Yuan Y., He S.H., Bandara A.R., Hyde K.D., Malysheva V.F., Li D.W., Dai Y.C. Global diversity and taxonomy of the 315 Auricularia auricula-judae complex (Auriculariales, Basidiomycota). Mycol. Prog. 2015, 14, 1–16. DOI: 10.1007/s11557-015-1113-4 316
In a paper 14: focusing on Global diversity of Auricularia, the concatenated ITS+nLSU and ITS+nLSU+rpb1+rpb2 dataset were used.
„ Most of the 31 Auricularia species formed monophyletic lineages with high support, and several species including A. heimuer and A. submesenterica didn’t form monophyletic lineages, however, these two species formed two distinct lineages with high support in the phylogeny based on the concatenated ITS+nLSU+rpb1+rpb2 dataset.“
- Wu F., Tohtirjap A., Fan L.F., Zhou L.W., Alvarenga R.L.M., Gibertoni T.B., Dai Y.C. Global diversity and updated phylogeny 317of Auricularia (Auriculariales, Basidiomycota). J. Fungi 2021, 7, 933. DOI: 10.3390/jof7110933
In paper 31:
Phylogenetic analysis by combined ITS, nLSU, and rpb2 sequences was reported.
„In addition, the rpb2 sequences of A. brasiliana were not successfully amplified, but an
analysis by sequences of ITS and LSU had a similar result as by ITS+nLSU+rpb2 sequences“.
Wu F., Yuan Y., Rivoire B., Dai Y.C. Phylogeny and diversity of the Auricularia mesenterica (Auriculariales, Basidiomycota) 350
complex. Mycol. Progress. 2015, 14, 1‒9. DOI: 10.1007/s11557-015-1065-8
Also in paper12: ITS + nLSU are used.
Wu F., Yuan Y., Malysheva V.F., Du P., Dai Y.C. Species clarification of the most important and cultivated Auricularia mushroom 313
“Heimuer”: evidence from morphological and molecular data. Phytotaxa 2014, 186, 241–253. DOI:10.11646/phytotaxa.186.5.1.
Complementing the results with another region does not require a lot of time or labor. However, I leave the decision regarding the acceptance to the article editor, but I insist that another region needs to be added.

Author Response
Dear reviewer,
we understand your recommendation to add another region and enhance reliability of results. As you write different regions are used with ITS, however, have different use among fungi groups. More regions were used in other research about Auricularia too, however, their benefit was not crucial. In the cited paper 13, there is presented phylogeny tree based only on ITS, with clear results for species discrimination. It is not significantly different against presented tree there which is based on more regions (but with ITS too) in species resolution.
We used ITS as the suitable and confirmed region for resolution of species in Auricularia. There is not stated that only ITS is sufficient where we added three citations 13,14,31 in discussion. We write that our described species is clearly supported by ITS. Our new Auricularia is sufficiently different from others as we can see in molecular analysis. It is not included in bad resolution complex species; we have 100 % support for separated clade (confirmed by morphological features). Hence, we do not need other regions.
To obtain more molecular data (more regions) may be easy if you have easily available molecular laboratory. We are not able to ensure it in several days. How many regions are sufficient for species resolution in Auricularia? We are not against opinion that other regions of DNA enhance reliability of new species. However, we stated that ITS region is sufficient. We base our results about new species at the experiences with Auricularia, too.
Reviewer 2 Report
Dear authors, you have changed the text and the improvements you have done present your results in a clear form. I still believe you should use all of the previous suggestions to have a better form of your manuscript.
Author Response
Dear reviewer,
thank you for your comment, other specific improving may be resolve with editor.